# Altered Pharmacokinetics of Ropivacaine in Patients Undergoing Laparoscopic Major Hepatectomy

**DOI:** 10.3390/pharmaceutics17030386

**Published:** 2025-03-18

**Authors:** Jun Zhang, Hongyuan Lv, Jiliang Shen, Zhichao Ai, Minjun Liu, Xiaorui Liu, Tieshuai Liu, Bo Shen, Hong Yu, Xin Yu

**Affiliations:** 1Department of Anesthesiology, Sir Run Run Shaw Hospital, School of Medicine, Zhejiang University, Hangzhou 310058, China; 3316088@zju.edu.cn (J.Z.); 22218685@zju.edu.cn (H.L.); 22318680@zju.edu.cn (Z.A.); 3415067@zju.edu.cn (T.L.); 2Department of General Surgery, Sir Run Run Shaw Hospital, School of Medicine, Zhejiang University, Hangzhou 310058, China; 3416233@zju.edu.cn (J.S.); simpleshen@zju.edu.cn (B.S.); blueyu000@zju.edu.cn (H.Y.); 3Department of Nursing Education, Sir Run Run Shaw Hospital, School of Medicine, Zhejiang University, Hangzhou 310058, China; liumj@srrsh.com; 4Department of Radiology, Sir Run Run Shaw Hospital, School of Medicine, Zhejiang University, Hangzhou 310058, China; lxr7871@126.com; 5Provincial Key Laboratory of Precise Diagnosis and Treatment of Abdominal Infection, Sir Run Run Shaw Hospital, School of Medicine, Zhejiang University, Hangzhou 310058, China

**Keywords:** alpha-1 acid glycoprotein, bilateral dual transversus abdominis plane block, laparoscopic major hepatectomy, local anesthetic toxicity, ropivacaine

## Abstract

**Background/Objectives:** Ropivacaine is primarily metabolized by the liver. High doses of ropivacaine, combined with altered pharmacokinetics due to hepatectomy, raise concerns about potential drug toxicity. We investigated the impact of LMH (laparoscopic major hepatectomy) on the pharmacokinetics of high-dose ropivacaine. **Methods:** Ten patients undergoing LMH received a BD-TAP (bilateral dual transversus abdominis plane) block with a high dose of ropivacaine (3 mg·kg^−1^ in 60 mL). Plasma concentrations of total and free ropivacaine and AAG (alpha-1 acid glycoprotein) levels were measured. Liver volumes were calculated using three-dimensional liver reconstruction technology. **Results:** The peak total ropivacaine concentration occurred 45 min after the block, reaching 2031.5 (876.0) ng·mL^−1^, with a tendency to exceed the toxicity threshold in patients with a CFLV (cut functional liver volume) exceeding 199.24 mL or a CFLV/TFLV (total functional liver volume) ratio surpassing 18.61%. The peak free ropivacaine concentration, 111.5 (31.3) ng·mL^−1^, was observed 90 min after the block, potentially exceeding the toxicity threshold when CFLV exceeded 452.33 mL or the CFLV/TFLV ratio was greater than 42.16%. Plasma AAG levels increased approximately 1.5 times within 24 h, from 1519.7 (422.6) μg·mL^−1^ preoperatively to 2253.6 (460.4) μg·mL^−1^ postoperatively, effectively reducing the toxicity risk associated with free ropivacaine. **Conclusions:** Preoperative administration of high-dose ropivacaine can be safely utilized in patients undergoing major hepatectomy. The increased plasma AAG concentration due to surgical stress reduces free ropivacaine levels, enhancing patient tolerance to the drug. The CFLV and CFLV/TFLV ratio may be supplementary indicators for predicting ropivacaine toxicity.

## 1. Introduction

Nerve blockade techniques have become a crucial component of multimodal opioid-sparing anesthesia, offering effective pain management [1]. Among the local anesthetics used in nerve blocks, ropivacaine is favored for its relatively low systemic toxicity compared to other long-acting anesthetics [2]. The liver is the primary organ responsible for the metabolism of ropivacaine, and clinical studies have shown that its distribution and elimination are altered in chronic end-stage liver disease [3]. This is largely because the cytochrome P450 enzymes responsible for ropivacaine metabolism are predominantly located in the hepatic lobule [4].

Furthermore, ropivacaine exhibits a high binding affinity to AAG (alpha-1 acid glycoprotein) in plasma [5]. AAG is an acute-phase protein mainly synthesized by the liver, and its plasma concentrations can increase by 3–5 times during surgery, cancer, trauma, infection, and inflammation [6,7]. The BD-TAP (bilateral dual transversus abdominis plane) block is an advanced regional anesthesia technique that involves injecting local anesthetic of the upper intercostal and lateral lower classical TAP (CL-TAP) compartments to anesthetize the upper (Th6–Th9) and the lower (Th10–L1) abdominal wall. This technique has shown effective postoperative analgesia after major abdominal surgeries, including laparoscopic hepatectomy [8,9]. It usually requires a high dose of local anesthetic to achieve broader and more effective analgesia. The fluctuations in plasma AAG concentrations due to surgical stimuli can significantly influence the binding rate of ropivacaine. Additionally, a CFLV (cut in functional liver volume) may impair the liver’s ability to synthesize AAG, potentially increasing patients’ susceptibility to drug toxicity.

Given these concerns, further research is necessary to understand how hepatectomy, through its effects on plasma AAG concentrations, influences the systemic safety of ropivacaine. Previous studies, such as Ollier et al., have explored the population pharmacokinetics of ropivacaine after TAP block in patients undergoing liver resection [10]. They found that although liver resections reduced the free ropivacaine clearance, the ropivacaine pharmacokinetic profile remained within the safe range. However, studies predicting ropivacaine toxicity in hepatectomy patients are lacking. This study aimed to evaluate the impact of laparoscopic major hepatectomy on the pharmacokinetics of high-dose ropivacaine, and especially investigate the role of AAG and liver volumetry in predicting ropivacaine toxicity.

## 2. Material and Methods

### 2.1. Study Design and Ethics Statement

This study was approved by the Ethics Committee of Sir Run Run Shaw Hospital, affiliated with Zhejiang University School of Medicine on 12 July 2017 (approval number: 201709501). Written informed consent was obtained from all participants. Additionally, the trial was registered prior to patient enrollment in the Chinese Clinical Trial Registry (ChiCTR-ONC-17012245, Principal investigator: Xin Yu, Date of registration: 4 August 2017). All procedures adhered to the provisions of the Declaration of Helsinki.

### 2.2. Participants

Between 1 March 2021, and 31 August 2021, consecutive patients undergoing LMH (laparoscopic major hepatectomy) for liver conditions, including liver stones, hemangioma, or liver cancer, were recruited. Exclusion criteria were: (1) a history of allergy to ropivacaine, (2) known tolerance to opioid analgesics, (3) inability to use patient-controlled intravenous analgesia, (4) need for laparotomy, (5) presence of mental disorders, and (6) BMI (body mass index) ≥ 40. All surgeries were performed by the same surgical team at Sir Run Run Shaw Hospital before BD-TAP Block.

### 2.3. Ultrasound-Guided BD-TAP Block

The ultrasound-guided BD-TAP block was performed by a trained anesthesiologist using a SonoSite SII ultrasound system, Boot version (52.80.111.013) (FUJIFILM SonoSite, Inc., Bothell, WA, USA) equipped with a high-frequency linear transducer (HFL38xi/13-6 MHz). An 80 mm regional block needle was inserted in a medial-to-lateral direction with an in-plane approach. Ropivacaine (Naropin, AstraZeneca, Gothenburg, Sweden) was administered at a dose of 3 mg·kg^−1^, diluted to a total volume of 60 mL, with 15 mL injected at each site [9].

### 2.4. Collection and Processing of Blood Samples

Blood samples (5 mL each) were drawn from the central vein preoperatively and at 15, 30, 45, 60, 90, 120, and 240 min post-block, as well as 24 h post-block. Samples were stored temporarily at 4 °C and centrifuged at 1100× *g* for 15 min within 2 h of collection. The plasma was then stored at −80 °C until analysis. Free and total ropivacaine concentrations in the post-block plasma samples were measured using LC–MS/MS (liquid chromatography–tandem mass spectrometry) [11]. AAG concentrations were determined using a double-antibody sandwich ELISA (enzyme-linked immunosorbent assay) kit (Shanghai Jianglai Industrial Co., Ltd., Shanghai, China).

### 2.5. Liver Volumetry

All participants underwent continuous transverse contrast-enhanced CT (computed tomography) of the abdomen before and after surgery, with a scanning interval of 0.5 cm. Liver volumetry, including TLV (total liver volume), DLV (diseased liver volume), and RLV (remnant liver volume), was calculated by two researchers using the liver 3D reconstruction software (Sensecare Liver, SenseTime (https://link.bi.sensetime.com/), Hong Kong, China) based on preoperative and postoperative CT scans. The DLV was considered an index of non-functional liver parenchyma. TFLV (Total functional liver volume) and CFLV were defined as TLV minus DLV and TFLV minus RLV, respectively. The ratio of RLV to TFLV or CFLV to TFLV (expressed as a percentage of TFLV) was also calculated.

### 2.6. Statistics

Statistical analyses were conducted using IBM SPSS 22 and GraphPad Prism 8. Data following a normal distribution were presented as mean (SD, standard deviation), and categorical variables were displayed by frequency (percentages). A simple linear regression analysis was performed to examine the relationship between liver volume and plasma ropivacaine concentration. A *p*-value of <0.05 was considered statistically significant.

## 3. Results

Ten patients who underwent LMH were enrolled in this study. No patients were excluded, and all completed the study as per the protocol. The clinical characteristics, surgical procedures, and ropivacaine doses for all patients are summarized in Table 1. Two patients had a decreased albumin value. Liver cirrhosis was found in three patients. All patients had a Child–Pugh score of A.

### 3.1. Plasma Ropivacaine Concentration

The pharmacokinetic parameters of total and free plasma ropivacaine concentrations are shown in Table 2, and the plasma concentration–time curves are presented in Figure 1. The mean peak total ropivacaine concentration in plasma was 2031.5 (876.0) ng·mL^−1^, occurring 45 min post-injection. In one case, the maximum individual concentration reached 3825.0 ng·mL^−1^, which occurred 30 min post-injection. The mean (SD) peak free plasma ropivacaine concentration occurred 1.5 h after ropivacaine administration, reaching an average of 111.5 (31.3) ng·mL^−1^. The highest individual concentration was 168.4 ng·mL^−1^, observed at 90 min. Individual total and free plasma ropivacaine concentrations are shown in Appendix A.

### 3.2. Free Ropivacaine Fraction

The free ropivacaine fraction is presented in Appendix A. Following the BD-TAP block, the mean (SD) peak free ropivacaine fraction was 7.1% (2.2), observed 1.5 h post-injection. Figure 2a illustrates the time curve for the total ropivacaine fraction within the first 24 h post-dose.

### 3.3. Plasma AAG Concentration

A significant increase in postoperative plasma AAG concentration was observed 24 h after the BD-TAP block, compared with preoperative levels (2253.6 (460.4) μg·mL^−1^ vs. 1519.7 (422.6) μg·mL^−1^, *p* = 0.002). Postoperative AAG concentrations surged to 1.5 times the preoperative level. The plasma AAG concentration–time curve is shown in Figure 2b. Plasma AAG concentrations for all patients are shown in Appendix A.

### 3.4. Liver Volumetry

Liver volume was calculated using Sensecare 3D reconstruction software based on enhanced abdominal CT scans preoperatively and postoperatively. Liver volume parameters and maximum ropivacaine concentrations for all patients are shown in Table 3.

### 3.5. Linear Regression Analysis

Over 30% of patients exhibited total ropivacaine concentrations exceeding the central nervous system toxicity threshold. A simple linear model between peak ropivacaine concentration and variables including weight, BMI, TLV, TFLV, RLV, CFLV, and CFLV/TFLV suggested that CFLV and CFLV/TFLV may be influencing factors leading to peak total ropivacaine concentrations exceeding the toxicity threshold (Figure 3).

## 4. Discussion

This study is the first to describe the pharmacokinetics of both free and total ropivacaine in patients undergoing major hepatectomy, incorporating 3D liver volumetry and AAG dynamics. The BD-TAP block is increasingly utilized as part of a multimodal approach to postoperative pain management [9,12,13,14]. Understanding the systemic absorption and disposition of ropivacaine following a BD-TAP block is crucial, particularly in patients with liver resection or impaired liver function, due to the associated risks of systemic toxicity. Our findings indicate that the operative stress-induced increase in plasma AAG concentrations enhances the binding of ropivacaine, thereby reducing the concentration of free ropivacaine and increasing the patient’s tolerance to ropivacaine toxicity. Despite the administration of high doses of ropivacaine for the BD-TAP block, no local anesthetic toxic reactions were observed in this cohort of LMH patients. Additionally, we found a correlation between the peak total and free ropivacaine concentrations and both CFLV and the ratio of CFLV/TFLV.

### 4.1. Plasma Ropivacaine Concentrations

Ropivacaine is a widely used local anesthetic, particularly in regional anesthesia, and is predominantly metabolized by cytochrome P450 enzymes in the liver [15]. Systemic absorption varies depending on the administration route and concentration. In healthy volunteers, the CNS (central nervous system) toxicity threshold for ropivacaine ranges from 1000 to 2200 ng·mL^−1^ [16,17]. However, the systemic toxic concentration can vary. For instance, a peak total plasma concentration (Cmax) of 3610 ng·mL^−1^ was reported without adverse reactions following the administration of 0.75% ropivacaine (17 mL) during ilioinguinal-iliohypogastric blocks after inguinal hernia repair [18]. Another study observed a mean peak concentration of 2540 (750) ng·mL^−1^, 30 min after a CL-TAP (classic TAP) block in patients undergoing gynecological surgery, using 3 mg·kg^−1^ of ropivacaine diluted to 40 mL (0.5% ropivacaine) [19]. Similarly, in cesarean section patients receiving a CL-TAP block with 2.5 mg·kg^−1^ of ropivacaine diluted to 40 mL (0.5% ropivacaine), the peak concentration was 1820 (690) ng·mL^−1^, 30 min post-injection [20].

In our study, the mean peak total ropivacaine concentration was 2031.5 (876.0) ng·mL^−1^, occurring 45 min post-injection with 3 mg·kg^−1^ of ropivacaine diluted to 60 mL (0.31% ropivacaine). The peak plasma concentrations of ropivacaine, a critical marker for potential CNS or cardiac toxicity, were dose-dependent, with the highest concentrations observed in the group receiving the highest dose. Notably, we found that in 30% of patients, total ropivacaine concentrations exceeded the CNS toxicity threshold of 2200 (800) ng·mL^−1^ without any evident signs of toxicity [17]. However, it is important to acknowledge that signs of CNS toxicity could have been missed, as most patients were still emerging from general anesthesia at the time of peak plasma concentrations. In a related study, only 25% (3/12) of patients exhibited toxic symptoms despite total ropivacaine concentrations surpassing the toxicity threshold [20]. Although total ropivacaine concentrations in 30% of our patients exceeded the CNS toxicity threshold, free ropivacaine concentrations remained within normal limits in 90% of the patients.

### 4.2. Is the Toxicity Threshold of Total Ropivacaine Concentration of Any Guiding Value?

The free plasma concentration of a drug is more closely related to its systemic pharmacodynamic effects and toxicity than the total concentration. For ropivacaine, the CNS toxicity threshold of free drug concentration has been reported as 150 (80) ng·mL^−1^ [17]. However, this value is primarily valid in patients without hepatic insufficiency. In our study, we observed a mean (SD) peak free ropivacaine concentration of 111.5 (31.3) ng·mL^−1^ at 1.5 h post-administration. The highest free ropivacaine concentration recorded was 168.4 ng·mL^−1^, yet no instances of seizures or cardiovascular instability were reported in patients undergoing LMH.

Ropivacaine is a basic drug with an approximately 94–95% binding ratio to AAG. Any change in the plasma concentration of this acute-phase protein can significantly impact the drug’s binding capacity, potentially influencing systemic toxicity [15,21]. AAG is synthesized by the liver, and its levels can vary based on the etiology and severity of liver dysfunction. In situations involving surgery, trauma, infection, or inflammation, AAG concentrations can increase by 3–5 times [6,7]. In healthy adults, the plasma AAG concentration typically ranges from 500 to 1000 µg·mL^−1^ [22]. In our study, we found that the mean preoperative AAG concentration was 1519.7 µg·mL^−1^, which is slightly higher than in healthy adults. This concentration increased by 1.5 times within 24 h postoperation.

Since AAG and drugs compete for the same binding sites, fluctuations in plasma AAG concentration can greatly influence the free drug concentration [23,24]. Our findings confirmed that elevated plasma AAG levels increase the binding of ropivacaine, thereby reducing its free concentration and enhancing the body’s tolerance to ropivacaine toxicity, even in patients who underwent LMH (Figure 1d, Figure 2 and Figure 3).

### 4.3. Do We Need to Adjust the Dosage of Ropivacaine in an Operative Setting? Especially in Patients Undergoing Hepatectomy?

The increase in plasma AAG concentration due to surgical stress, as observed in both previous studies and the current one, enhances the binding capacity of ropivacaine, thereby reducing its free fraction. This reduction in free ropivacaine decreases the risk of adverse effects from the local anesthetic block, suggesting that surgical patients might tolerate larger doses of local anesthetic. However, patients undergoing major hepatectomy or those with liver dysfunction face reduced intrinsic drug clearance and diminished synthetic capacity. As shown in Figure 2b, 45 min post-injection, AAG concentration began to decrease, reaching its lowest level at 2 h post-injection. This decline might be attributed to the reduced capacity to synthesize AAG following hepatectomy.

Given these opposing mechanisms affecting AAG concentration, local anesthetic infiltration in patients undergoing major hepatectomy is a significant concern. While “major hepatectomy” is an anatomical term, it does not fully reflect the functional reserve of the liver. This study is the first to employ liver volumetry to examine the relationship between the extent of liver resection and ropivacaine concentration.

We found that both CFLV and the CFLV/TFLV ratio had a linear relationship with peak ropivacaine concentration. When CFLV exceeded 199.24 mL or the CFLV/TFLV ratio surpassed 18.61%, the peak total ropivacaine concentration exceeded the toxicity threshold (Figure 3a,b). For the free ropivacaine concentration, the safety thresholds for CFLV and CFLV/TFLV were 452.33 mL and 42.16%, respectively (Figure 3c,d). Therefore, in patients undergoing LMH with a BD-TAP block, if CFLV exceeds 452.33 mL or the CFLV/TFLV ratio is greater than 42.16%, the ropivacaine dosage should be reduced. In hepatectomy models, CFLV/TFLV appears to be a more accurate predictor of drug metabolism than anatomical liver volume alone.

The total ropivacaine fraction–time curve highlights AAG as a critical protective factor against ropivacaine toxicity. Even slight fluctuations in plasma AAG concentration can significantly impact free ropivacaine levels. Enhanced AAG synthesis can improve tolerance to ropivacaine toxicity. In clinical practice, drug dosages are typically based on the patient’s body weight. Although weight-based dosing is generally effective, it may not be suitable for patients undergoing major surgery, particularly those involving the liver. To prevent ropivacaine toxicity, we recommend close monitoring of neurological and cardiovascular symptoms in patients postoperatively, especially at the time of peak ropivacaine concentration (typically 45 min to 2 h after administration). Additionally, regular monitoring of plasma AAG levels and ropivacaine concentrations, particularly in patients with higher CFLV or CFLV/TFLV ratios, may help identify potential toxicity risks at an early stage.

There are some limitations of our study. First, the sample size of this study was relatively small, with only ten patients included, which limits the statistical power. Future studies should consider increasing the sample size and including control groups (e.g., patients undergoing minor hepatectomy or no liver resection) to further validate our findings and provide a stronger comparative framework. Second, future studies should closely monitor neurological and cardiovascular symptoms in patients, especially at the time of peak ropivacaine concentration, to ensure that potential toxicity reactions can be promptly identified and managed. Third, follow-up studies should focus on practical dosage modifications when CFLV exceeds 452.33 mL or the CFLV/TFLV ratio surpasses 42.16%.

## 5. Conclusions

This study demonstrates that ultrasound-guided BD-TAP block with a high dose of ropivacaine can be both safe and effective for postoperative analgesia in patients undergoing LMH. The increase in AAG due to surgery results in greater binding of the local anesthetic, thereby lowering its free concentration and enhancing tolerance to CNS toxicity. However, caution is advised when CFLV exceeds 452.33 mL or the CFLV/TFLV ratio surpasses 42.16%, as the peak free ropivacaine concentration may exceed the toxicity threshold. Close monitoring is recommended when administering large doses of ropivacaine in patients undergoing major hepatectomy.

## Figures and Tables

**Figure 1 pharmaceutics-17-00386-f001:**
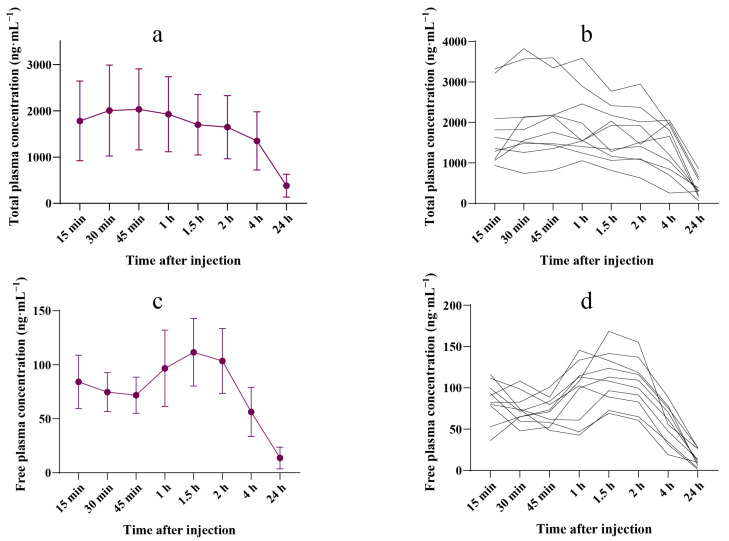
Ropivacaine concentration–time curve. (**a**) Mean (SD) plasma concentration–time profiles of total ropivacaine during the 24 h after the injection (*n* = 10). (**b**) Individual plasma concentration–time profiles of total ropivacaine during the 24 h after the injection (*n* = 10). (**c**) Mean (SD) plasma concentration–time profiles of free ropivacaine during the 24 h following the injection (*n* = 10). (**d**) Individual plasma concentration–time profiles of free ropivacaine during the 24 h following the injection (*n* = 10).

**Figure 2 pharmaceutics-17-00386-f002:**
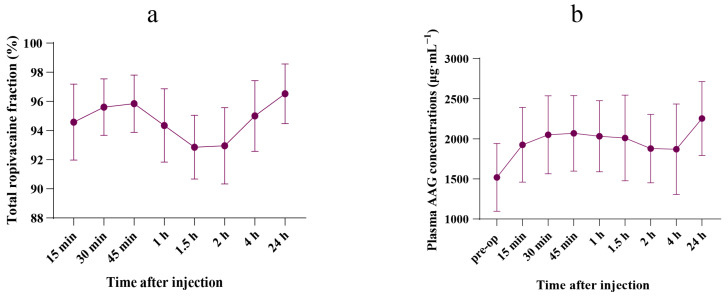
Concentrations–time curve. (**a**) Mean (SD) fraction–time curve of total ropivacaine during the 24 h after the injection (*n* = 10). (**b**) Mean (SD) plasma concentrations–time curve of AAG during the 24 h after the injection (*n* = 10).

**Figure 3 pharmaceutics-17-00386-f003:**
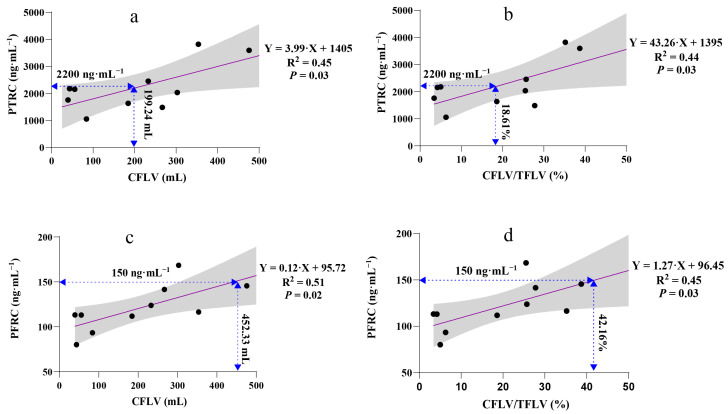
Linear regression analysis of the associations between peak ropivacaine concentration and variables. A 95% confidence interval (grey band) can express uncertainty in a linear regression relationship. (**a**) Linear regression analysis of PTRC and CFLV, F (1,8) = 6.67 and adjusted R^2^ = 0.39; (**b**) Linear regression analysis of PTRC and CFLV/TFLV%, F (1,8) = 6.18 and adjusted R^2^ = 0.37; (**c**) Linear regression analysis of PFRC and CFLV, F (1,8) = 8.21 and adjusted R^2^ = 0.45; (**d**) Linear regression analysis of PFRC and CFLV/TFLV%, F (1,8) = 6.48 and adjusted R^2^ = 0.38. PTRC, peak total ropivacaine concentration; PFRC, peak free ropivacaine concentration; CFLV, cut functional liver volume; TFLV, total functional liver volume.

**Table 1 pharmaceutics-17-00386-t001:** Clinical characteristics and operation procedures of the ten patients.

	1	2	3	4	5	6	7	8	9	10	Mean (SD)/*n* (%)
General background											
Female (yes/no)	yes	yes	yes	no	yes	no	yes	no	yes	yes	7 (70)
Age (year)	52	56	71	61	62	35	58	73	31	67	56.6 (14.0)
Height (cm)	162	152	150	171	160	182	161	181	163	161	164.3 (10.7)
Weight (kg)	70	55	52	67	65	71	55	78	50	62	62.5 (9.2)
BMI (kg·m^−2^)	26	23	23	22	25	21	21	23	18	23	22.5 (2.2)
Ropivacaine (mg)	210	165	156	201	195	213	165	234	150	186	187.5 (27.8)
Ropivacaine (%)	0.35	0.27	0.26	0.33	0.32	0.35	0.27	0.39	0.25	0.31	0.31 (0.04)
Blood examination											
PT (s)	12.4	12.2	13.0	13.5	12.2	13.0	12.6	13.5	13.3	13.1	12.8 (0.5)
ALT (U·L^−1^)	13	49	7	15	23	43	10	22	10	45	23.7 (16)
TB (μmol·L^−1^)	14.5	12.3	8.3	17.3	10.7	11.7	12.2	12.3	11	21.3	13.1 (3.7)
Alb (g·L^−1^)	47.9	48.4	33.2 *	40.5	44.3	46.7	36.6 *	40.3	44.5	44.8	42.7 (4.9)
Liver functional reserve											
liver cirrhosis (yes/no)	no	no	no	yes	yes	yes	no	no	no	no	3 (30)
Child–Pugh score	A	A	A	A	A	A	A	A	A	A	
Surgical procedure											
Surgical indications	1	2	3	3	3	3	2	3	2	3	
Operative time (min)	270	135	270	350	210	225	165	200	310	300	243.5 (68.0)
Blood loss (mL)	500	50	300	1000	200	200	200	200	200	300	315.0 (266.7)
Blood transfusion (U)	0	0	0	4	0	0	0	0	0	0	

**Notes:** Data are presented as counts (percentage) or means (SD), as appropriate. 1: Hepatic hemangioma; 2: Hepatolithiasis; 3: Hepatoma. * Values above or below the normal range (normal range: prothrombin time 11.5–14.5 s, alanine transaminase 9–50 U·L^−1^, total bilirubin 0–26 μmol·L^−1^, albumin 40–55 g·L^−1^). **Abbreviations:** PT, prothrombin time; ALT, alanine transaminase; TB, total bilirubin; Alb, albumin.

**Table 2 pharmaceutics-17-00386-t002:** Pharmacokinetic parameters of plasma ropivacaine of the ten patients.

Parameter	Unit	1	2	3	4	5	6	7	8	9	10	Mean (SD)
Total plasma ropivacaine												
AUC_(0–t)_	μg·L^−1^·h	7960.2	24,639.6	36,138.1	32,950.3	16,734.9	37,106.9	11,903.6	21,654.8	29,790.6	18,525.9	23,740.5 (10,134.3)
AUC_(0–∞)_	μg·L^−1^·h	7960.3	26,211.5	44,336.8	45,596.7	17,268.6	57,505.0	12,468.1	25,471.0	31,608.6	22,918.0	29,134.5 (15,778.5)
MRT_(0–t)_	h	10.6	4.72	6.57	7.64	8.00	8.22	4.37	6.24	4.63	6.67	6.77 (1.95)
MRT_(0–∞)_	h	10.6	6.48	12.4	17.5	8.72	22.2	5.62	10.7	6.26	12.8	11.3 (5.3)
t_1/2z_	h	1.50	7.03	9.72	13.3	5.48	16.4	5.62	8.47	6.28	10.1	8.39 (4.25)
T_max_	h	1.00	0.750	0.500	1.50	0.750	1.00	0.250	0.750	0.750	0.500	0.775 (0.343)
Vz	L·kg^−1^	0.817	1.16	0.949	1.26	1.38	1.24	1.95	1.44	0.860	1.92	1.30 (0.40)
CLz	L·h^−1^·kg^−1^	0.377	0.114	0.068	0.066	0.174	0.052	0.241	0.118	0.095	0.131	0.144 (0.099)
C_max_	μg·L^−1^	1055.0	2178.5	3825.0	2034.0	1761.8	2457.2	1635.0	2157.8	3597.9	1485.2	2218.7 (883.2)
Free plasma ropivacaine												
AUC_(0–t)_	μg·L^−1^·h	489.9	597.0	904.9	1240.6	1084.6	1404.1	618.5	1142.9	1262.5	1606.4	1035.1 (372.7)
AUC_(0–∞)_	μg·L^−1^·h	606.9	619.2	1059.7	1564.4	1206.8	1829.6	625.4	1223.9	1306.4	1974.8	1201.7 (492.2)
MRT_(0–t)_	h	7.07	4.23	6.27	7.54	5.52	7.46	3.31	4.99	4.26	6.81	5.75 (1.51)
MRT_(0–∞)_	h	12.6	5.20	10.5	13.5	8.43	14.9	3.59	6.84	5.16	12.6	9.33 (4.01)
t_1/2z_	h	8.20	4.98	7.95	8.50	7.12	10.6	3.68	6.16	5.07	9.42	7.16 (2.17)
T_max_	h	0.25	0.25	0.25	1.50	1.00	1.50	0.25	1.50	1.00	1.50	0.90 (0.60)
Vz	L·kg^−1^	58.5	34.8	32.5	23.5	25.5	25.0	25.5	21.8	16.8	20.7	28.5 (11.8)
CLz	L·h^−1^·kg^−1^	4.94	4.84	2.83	1.92	2.49	1.64	4.80	2.45	2.30	1.52	2.97 (1.36)
C_max_	μg·L^−1^	93.3	80.1	116.4	168.4	113.2	123.8	111.8	113.1	145.6	141.5	120.7 (25.7)

**Notes:** Data are presented as mean (SD), as appropriate. **Abbreviations:** AUC_(0–t)_: area under the plasma concentration–time curve from time 0 to the time of last measurable concentration; AUC_(0–∞)_: area under the plasma concentration–time curve from time 0 to infinity; MRT_(0–t)_: mean residence time from time 0 to the time of last measurable concentration; MRT_(0–∞)_: mean residence time from time 0 to infinity; t_1/2z_: plasma terminal elimination half-life; T_max_: time to peak concentration; Vz: apparent volume of distribution; CLz: apparent total plasma clearance; C_max_: peak concentration.

**Table 3 pharmaceutics-17-00386-t003:** Liver volume parameters and maximum ropivacaine concentrations of the ten patients.

	TLV (mL)	DLV (mL)	TFLV (mL)	RLV (mL)	CFLV (mL)	CFLV/TFLV (%)	PTRC (ng·mL^−1^)	PFRC (ng·mL^−1^)
1	1882.9	535.8	1347.1	1263.2	83.9	6.2	1055.0	93.3
2	871.4	5.1	866.3	822.7	43.6	5.0	2178.5	80.1
3	1074.5	67.6	1006.9	653.2	353.7	35.1	3825.0	116.4
4	1547.2	357.7	1189.5	886.7	302.8	25.5	2034.0	168.4
5	1339.4	171.9	1167.5	1127.9	39.6	3.4	1761.8	113.2
6	1136.3	228.4	907.9	675.0	232.9	25.7	2457.2	123.8
7	1010.3	20.5	989.8	805.1	184.7	18.7	1635.0	111.8
8	1443.3	114.3	1329.0	1273.2	55.8	4.2	2157.8	113.1
9	1247.6	14.7	1232.9	756.9	476.0	38.6	3597.9	145.6
10	982.4	21.0	961.4	694.7	266.7	27.7	1485.2	141.5
Mean (SD)	1253.5 (307.4)	153.7 (176.1)	1099.8 (174.9)	895.9 (238.6)	204.0 (149.2)	19.0 (13.5)	2218.7 (883.2)	120.7 (25.7)

**Notes:** Data are presented as means (SD), as appropriate. **Abbreviations:** TLV: total liver volume; DLV: diseased liver volume; TFLV: total functional liver volume; RLV: remnant liver volume; CFLV: cut functional liver volume; CFLV/TFLV: cut functional liver volume (CFLV) to total functional liver volume (TFLV) ratio; PTRC: peak total ropivacaine concentration; PFRC: peak free ropivacaine concentration.

## Data Availability

The data that support the findings of this study are available from the corresponding author upon reasonable request.

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
