# Peer review of "Altered Pharmacokinetics of Ropivacaine in Patients Undergoing Laparoscopic Major Hepatectomy"

_pharmaceutics, 2025, doi:10.3390/pharmaceutics17030386_

Round 1
Reviewer 1 Report
Comments and Suggestions for Authors
This manuscript presents an important and clinically relevant study on the altered pharmacokinetics of ropivacaine in patients undergoing laparoscopic major hepatectomy (LMH). The research is well-designed, using LC-MS/MS and 3D liver volumetry to assess changes in ropivacaine metabolism post-surgery. Preoperative administration of high-dose ropivacaine can be safely used in LMH patients. Increased plasma AAG levels due to surgical stress enhance drug tolerance. CFLV and CFLV/TFLV ratios serve as potential indicators for predicting ropivacaine toxicity.
The findings provide valuable insights into the role of alpha-1 acid glycoprotein (AAG) in mitigating toxicity risks, making it a great contribution to the field of anesthesiology and hepatectomy pharmacology, and it offers an important understanding of ropivacaine safety.
However, there are some areas that could be improved to enhance the study’s impact and applicability in the following areas.
- The current study includes only 10 patients, which limits statistical power. Adding a control group (e.g., patients undergoing minor hepatectomy or no liver resection) would provide a stronger comparative framework for understanding pharmacokinetic alterations.
- While pharmacokinetic parameters are well analyzed, clinical toxicity symptoms (e.g., neurological or cardiovascular effects) are not extensively reported. Including real-world adverse reactions would make the study more clinically actionable.
- The study identifies CFLV and CFLV/TFLV ratios as indicators of toxicity risk, but practical recommendations on dosage modifications based on these values would be beneficial for clinicians.
- While patient safety is addressed, further elaboration on monitoring protocols for toxicity prevention would be helpful for practical implications.
Here are some minor points.
- In Figure 1, the order of the figure caption is a,c,b,e,d,f. It must be in alphabetic order.
- The caption for Figure 3 a-d has only the results (R values) and lacks clear explanatory notes; what ratio is shown in each chart? It is helpful for readers to interpret them.
Author Response
Reviewer 1, Comment 1: The current study includes only 10 patients, which limits statistical power. Adding a control group (e.g., patients undergoing minor hepatectomy or no liver resection) would provide a stronger comparative framework for understanding pharmacokinetic alterations.
Response: Thank you for your insightful comment. We fully recognize that the small sample size and the absence of a control group are limitations of the study if conducted as a clinical study. The sample size calculation for this study was based on previous studies on pharmacokinetics (PMID: 22257580). In addition, considering that eight blood samples need to be collected, it may be a concern for many patients to decline inclusion. Therefore, 10 patients were finally included in this study in the hope of providing clinical implications using a pharmacokinetic model. Surprisingly, our study concluded that CFLV and CFLV/TFLV ratios may be adjunctive predictors of ropivacaine toxicity. This would be clinically instructive. Including a control group (e.g., patients undergoing minor hepatectomy or no liver resection) and expanding the sample size will help better understand the impact of hepatectomy on the pharmacokinetics of ropivacaine. We have explicitly mentioned this limitation in the Discussion section and recommend that future studies include more patients and control groups.
Reviewer 1, Comment 2:While pharmacokinetic parameters are well analyzed, clinical toxicity symptoms (e.g., neurological or cardiovascular effects) are not extensively reported. Including real-world adverse reactions would make the study more clinically actionable.
Response: Thank you for your feedback. We agree that reporting clinical toxicity symptoms is crucial for the clinical relevance of the study. In this study, although the total ropivacaine concentration exceeded the central nervous system (CNS) toxicity threshold in some patients, no obvious toxicity symptoms were observed. This may be related to the fact that patients were still under general anesthesia at the time of peak concentration. We have provided details on this in the Discussion sections and recommend that future studies closely monitor neurological and cardiovascular symptoms.
Reviewer 1, Comment 3: The study identifies CFLV and CFLV/TFLV ratios as indicators of toxicity risk, but practical recommendations on dosage modifications based on these values would be beneficial for clinicians.
Response: Thank you for your valuable feedback. What you are suggesting is perfectly in line with the clinical significance of what we are looking for in this study. With concerns about potential drug toxicity for high dose of ropivacaine in patients undergoing LMH, our study focused on studying the altered pharmacokinetics of ropivacaine in LMH, and exploring the relationship between ropivacaine concentration and functional liver volume and AAG. Interestingly, based on our findings, when CFLV exceeds 452.33 mL or the CFLV/TFLV ratio is greater than 42.16%, the peak ropivacaine concentration may exceed the toxicity threshold. We agree that providing dosage adjustment recommendations based on CFLV and CFLV/TFLV ratios would be beneficial for clinical practice. We recommend lowering the drug dose when these values are exceeded which was shown in the Discussion sections. However, in this small sample preliminary study, it is difficult to characterize the recommended reduction of a definitive dose of ropivacaine when CFLV exceeds 452.33 mL or when the CFLV/TFLV ratio is greater than 42.16%. This will be an issue for future studies to address, and we have expressed this in the study limitations.
Reviewer 1, Comment 4: While patient safety is addressed, further elaboration on monitoring protocols for toxicity prevention would be helpful for practical implications.
Response: Thank you for your feedback. We agree that further elaboration on monitoring protocols for toxicity prevention would enhance the clinical applicability of the study. We provide related recommendations on how to implement monitoring in clinical practice to ensure patient safety.
Reviewer 1, Comment 5: In Figure 1, the order of the figure caption is a,c,b,e,d,f. It must be in alphabetic order.
Response: Thank you for your feedback. We have reordered the figure caption in Figure 1.
Reviewer 1, Comment 6: The caption for Figure 3 a-d has only the results (R values) and lacks clear explanatory notes; what ratio is shown in each chart? It is helpful for readers to interpret them.
Response: Thank you for your feedback. We have provided clear explanatory notes in the caption for Figure 3.
Reviewer 2 Report
Comments and Suggestions for Authors
General comments
The authors measured ropivacaine and AAG concentrations after TAP block in patients undergoing hepatectomy. Although population pharmacokinetic model of ropivacaine after TAP block in patients undergoing liver resection is already published (PMID: 25557141), the authors also measured AAG concentrations. I believe that the study is well-done. I have several major concerns. First, the authors need to clearly state the novelty of the study and discuss the differences between the study and Ollier’s study (PMID: 25557141). Second, what is the bilateral dual TAP block? Third, there are too much data shown in tables.
Major concerns
#1. Why don’t the authors refer Ollier’s study (PMID: 25557141), which shows population pharmacokinetic model of ropivacaine after TAP block in patients undergoing liver resection?
The authors need to clearly state the novelty of the study and discuss the differences between the study and Ollier’s study.
#2. What is the bilateral dual TAP block? Ref. #8 does not show the bilateral dual TAP block. Please show the reference, which shows that the block is how delivered and widely used in hepatectomy.
#3. There are too much data shown in tables. Figures 1-b and 1-e should be removed. The data can be seen in Figures 1-c and 1f. Individual data should be shown in supplemental tables.
Minor comments
#1. The first sentence of discussion is not correct. Olliers already show the pharmacokinetics. Please change expressions.
#2. I do not think Figure 4 is necessary. Please remove Figure 4.
Author Response
Reviewer 2, Comment 1: Why don’t the authors refer Ollier’s study (PMID: 25557141), which shows population pharmacokinetic model of ropivacaine after TAP block in patients undergoing liver resection? The authors need to clearly state the novelty of the study and discuss the differences between the study and Ollier’s study.
Response: We thank for pointing out the need to reference Ollier’s study (PMID: 25557141) and to clearly state the novelty of our study. Ollier’s study developed a population pharmacokinetic model of ropivacaine after TAP block in patients undergoing liver resection, focusing on the general pharmacokinetics of ropivacaine in this population. In contrast, our study specifically investigates the impact of laparoscopic major hepatectomy (LMH) on the pharmacokinetics of high-dose ropivacaine(3 mg·kg⁻¹, diluted to a total volume of 60 mL,with 15 mL injected at each site preoperatively) with a particular focus on the role of alpha-1 acid glycoprotein (AAG) and functional liver volume (CFLV and CFLV/TFLV ratios) in predicting ropivacaine toxicity. This is a novel aspect not addressed in Ollier’s study. We will explicitly discuss these differences and highlight the novelty of our findings in the revised manuscript.
Reviewer 2, Comment 2: What is the bilateral dual TAP block? Ref. #8 does not show the bilateral dual TAP block. Please show the reference, which shows that the block is how delivered and widely used in hepatectomy.
Response: Thank you for your feedback. We have provided a more detailed explanation of the technique and cited appropriate references in the Introduction sections.
Reviewer 2, Comment 3: There are too much data shown in tables. Figures 1-b and 1-e should be removed. The data can be seen in Figures 1-c and 1f. Individual data should be shown in supplemental tables.
Response: Thank you for your feedback. We have removed Figures 1-b and 1-e, as the data can be adequately represented in Figures 1-c and 1-f.Besides, this was a small sample study and the statistical results are not fully representative of the pharmacokinetic profile of ropivacaine in LMH patient, so we have presented the individual data in the table.
Reviewer 2, Comment 4: The first sentence of discussion is not correct. Olliers already show the pharmacokinetics. Please change expressions.
Response: Thank you for your feedback. We have revised the sentence to reflect that Ollier’s study has already described the pharmacokinetics of ropivacaine in liver resection patients.
Reviewer 2, Comment 5: I do not think Figure 4 is necessary. Please remove Figure 4.
Response: Thank you for your feedback. We agree that Figure 4 (conceptual map) is redundant. Its key message—using CFLV/TFLV to guide dosing—is well described in the text.
Round 2
Reviewer 2 Report
Comments and Suggestions for Authors
The authors improved manuscript appropriately. I have no further comments.
Author Response
Comment 1:The authors improved manuscript appropriately. I have no further comments.
Response:
Dear Reviewer,
Thank you very much for your thorough review and valuable feedback on our manuscript. We are pleased to learn that the revised version of the paper has met your expectations and that no further modifications are required. Your comments have been instrumental in helping us improve the quality and rigor of our work.
We will ensure that the final version of the manuscript adheres to the highest standards of academic excellence, as per your suggestions. Once again, we deeply appreciate the time and effort you have dedicated to providing us with your expert guidance.
Sincerely,
Xin Yu M.D.
Department of Anaesthesiology, Sir Run Run Shaw Hospital, School of Medicine, Zhejiang University, Hangzhou 310016, China.
Provincial Key Laboratory of Precise Diagnosis and Treatment of Abdominal Infection, Sir Run Run Shaw Hospital, School of Medicine, Zhejiang University, Hangzhou, Zhejiang 310016, China
E-mails: xinxin_yu@zju.edu.cn